# Association between Dental Fear and Children’s Oral Health-Related Quality of Life

**DOI:** 10.3390/ijerph21091195

**Published:** 2024-09-10

**Authors:** Fahad Hegazi, Nada Alghamdi, Danah Alhajri, Lulwah Alabdulqader, Danah Alhammad, Lama Alshamrani, Sumit Bedi, Sonali Sharma

**Affiliations:** 1Preventive Dental Science Department, Imam Abdulrahman Bin Faisal University, Dammam 34212, Saudi Arabia; sbrajinder@iau.edu.sa; 2Collage of Dentistry, Imam Abdulrahman Bin Faisal University, Dammam 34212, Saudi Arabia; 2190001469@iau.edu.sa (N.A.); 2180004047@iau.edu.sa (D.A.); 2180001573@iau.edu.sa (L.A.); 2180001785@iau.edu.sa (D.A.); 2190003115@iau.edu.sa (L.A.); 3Biomedical Dental Sciences Department, Imam Abdulrahman Bin Faisal University, Dammam 34212, Saudi Arabia; svsharma@iau.edu.sa

**Keywords:** OHRQoL, COHIP-SF19, CFSS-DS, children, dental phobia, dental experience, parents

## Abstract

Objectives: This study aimed to examine the association between both parental dental fear and children’s dental fear and its impact on the Oral Health-Related Quality of Life (OHRQoL) of Saudi children in the Eastern Province of Saudi Arabia. Methods: Data on 93 individuals aged 7–12 years were collected using clinical examination and Arabic-validated questionnaires: the Child Oral Health Impact Profile-Short Form (COHIP-SF19), and the Children’s Fear Survey Schedule—Dental Subscale (CFSS-DS). Negative binomial regression analysis was performed to study the association between children’s fear and parental dental fear as well as the OHRQoL, while adjusting for certain confounders. Results: Overall, our multivariate analyses showed that children with high dental fear (CFSS-DS ≥ 38, *p* = 0.027) and a higher percentage of dental caries (*p* = 0.013) had a significantly lower OHRQoL after adjusting for clinical and socio-demographic factors. Further, parental fear of dentists was significantly associated with children’s fear of dentists (*p* < 0.001). Conclusions: Our findings demonstrate that children’s fear and dental caries are both associated with poorer OHRQoL. Further, parental dental fear about dentists was associated with children’s fear of dentists.

## 1. Introduction

Dental fear is a common concern among children and parents, significantly impacting oral health and overall well-being [1]. Among various factors related to an individual’s general well-being, oral health plays an integral role [2]. Health-Related Quality of Life (HRQoL) measures the impact of disease on daily functioning [3], and dental fear often leads to compromised oral hygiene practices and neglected dental care [4,5]. As a result, children experience a lower quality of life regarding oral health [4,5]. To assess this impact, multiple Oral Health-Related Quality of Life (OHRQoL) scales were developed [2].

OHRQoL is assessed using questionnaires that consider the functional, psychological, and social impacts of oral health [6]. Multiple OHRQoL questionnaires exist, tailored for different age groups [6,7,8]. For children, the Early Childhood Oral Health Impact Scale (ECOHIS) measures OHRQoL in ages 3–5 [9], while the Child Oral Health Impact Profile-Short Form (COHIP-SF19) assesses children aged 7–18 [6,7,8,10]. Factors associated with lower OHRQoL include age [11], gender [12], education level [4], family income [4], geographic location [6], body mass index (BMI) [13], dental caries [4], malocclusion [14], child’s fear [1], and parental dental fear [15,16]. Dental fear often leads to avoidance behaviors, resulting in a lack of regular dental care and delay in seeking necessary treatment [5]. Oral health remains a significant public health concern in Saudi Arabia, where studies have shown varying levels of awareness and inconsistent oral hygiene practices [17]. Moreover, research in the Eastern province of Saudi Arabia concluded that poor oral hygiene practices, lack of parental guidance and appropriate dental health knowledge with frequent exposure to cariogenic foods in addition to socio-demographics are the main risk factors for dental decay [18]. CFSS-DS is a widely used questionnaire [19]. It rates fear across 15 dental-related situations using a 5-point Likert scale [19]. Researchers often employ this scale to determine a child’s level of fear related to dental visits [19]. Further, factors such as age, gender, income, education, and dental caries are also associated with an increase in dental fear [20,21]. Recently, Diotaiuti et al. (2023) emphasized the importance of ensuring measurement invariance across genders in fear assessment tools, a factor that warrants further exploration in future studies on dental fear [22].

While we recognize that geographic location is linked to OHRQoL [6], it was crucial to investigate this association between children’s fear and OHRQoL within the population of Saudi Arabia. Previous studies conducted in Saudi Arabia did not adequately account for socio-demographic and clinical factors that could have acted as confounders and potentially influenced the study results. A previous study by Merdad L et al., did not study the influence of parental dental fear and the association between both parental dental fear and children’s dental fear and its impact on the Oral Health-Related Quality of Life (OHRQoL) of Saudi children [23]. Parental dental fear is one of the critical factors that often leads to avoidance behaviors, resulting in a lack of regular dental care and delay in seeking necessary treatment for the child patient. Our study aimed to examine this significant association. Thus, our primary aim was to examine the association between children’s fear levels and the OHRQoL in Saudi Arabia, while adjusting for certain confounders that could have adversely affected the association between these two variables. Our secondary aim was to study the association between parental dental fear and children’s fear while adjusting for certain confounders.

## 2. Materials and Methods

### 2.1. Ethical Considerations

The research ethical approval (IRB-2022-02-379) was obtained from the Deanship of Scientific Research, Imam Abdulrahman Bin Faisal University, Dammam. Participation in the study was voluntary, and participants were assured of the anonymity, privacy, and confidentiality of their data. The purpose, details, and expected benefits of the study were explained to all participants. Ethical guidelines expressed in the Declaration of Helsinki were followed during the conduct of the study.

This cross-sectional study recruited Saudi children between the ages of 7 and 12 years of age and the parents of these children. All parents who attended the IAU Dental Hospital between September 2022 and February 2023 are included. Further, the authors followed the STROBE checklist for this manuscript.

### 2.2. Sample Size

A sample size calculation was conducted to ensure adequate statistical power for the study. A power calculation was performed using Stata/SE16.1 (StataCorp, College Station, TX, USA). Assuming that the mean OHRQoL of fearful children and non-fearful children were 32.6 and 22.6 with a standard deviation of 16.2, a sample size of N = 85 was adequate to obtain a type 1 error rate of 5% and a power of 80%. We were able to collect a sample size of 93 [23].

The inclusion criteria involved: Saudi children; Arabic-speaking parentschildren between the ages of 7 and 12 yearsASA I/II children.

The exclusion criteria are:Non-Saudi childrenNon-Arabic speaking parentsASA III/IV childrenRefusal to give consent and assent for enrolment in the study.

### 2.3. Questionnaire Assessment

Informed consent was obtained from parents as part of our online questionnaire using Google Forms, which included inquiries concerning the child’s oral health-related quality of life. The informed consent was explained to the parents, and only after they had the opportunity to ask questions about the study and the consent itself, was the informed consent obtained. The questionnaire incorporated the Arabic-validated versions of the COHIP-SF19 and the CFSS-DS. (Appendix A) COHIP-19 is a shortened version of the Child Oral Health Impact Profile (COHIP), comprising 19 questions grouped into three categories: oral health (5 items), functional well-being (4 items), and socio-emotional well-being (10 items). The total score, ranging from 0 to 76, was calculated by summing scores for all 19 items [10]. The CFSS-DS comprises 15 items assessing fear responses to different stimuli. Responses are rated on a scale from 1 to 5, ranging from “not afraid at all” to “very much afraid”. The total score ranges from 15 to 75. Children scoring below 38 were categorized as non-fearful, and those scoring greater than or equal to 38 were categorized as fearful. The CFSS-DS is widely recognized as one of the most frequently utilized psychological scales for children [19]. Additionally, a section from a previous study consisting of five questions regarding the child’s age, parent’s age, income, education level, and smoking status was included (Appendix A) [1,10].

### 2.4. Clinical Examination

Two examiners were calibrated for conducting oral clinical exams. An inter-reliability test was performed with a Kappa score of 0.92. Participants’ weight was measured using a calibrated digital scale, height was measured using a stadiometer. Clinical examination was conducted using a sharp explorer, and a dental mirror. The oral examination recorded Angle’s molar classification, presence of crossbite, presence of an open-bite, midline shift, and decayed teeth. 

### 2.5. Defining Outcomes, Exposures, and Confounders

OHRQoL: This was measured and analyzed as a count variable using the COHIP-SF 19.

Children’s Dental Fear: Was measured using the CFSS-DS and analyzed as a count variable and as a binary variable (CFSS-DS ≥ 38; high fear) [24].

Parent’s fear of dentists: Measured using a 5-point Likert Scale. However, it was analyzed as a continuous variable [15].

Parent’s previous bad experience: Measured as a binary variable where they answered “yes/no” to “if they had ever had a previous bad experience with a dentist when they were a child” [15].

Parent’s comfort at the last visit to the dentist: Measured as a binary variable where they answered, “comfortable/anxious” [15].

Caries Percentage: The percentage of caries was calculated as the number of carious primary and permanent teeth divided by the total number of primary and permanent teeth.

Child’s age: Was measured as a continuous variable.

Child’s sex: Was measured as a binary variable (male/female).

Education: Was measured as a categorical variable (less than high school; high school; more than high school).

Family Income: Was measured as a categorical variable (low: <9000 Saudi Riyals (SAR); medium: 9000–12,000 SAR; high: >12,000 SAR) [25].

Parental Smoking: Was measured as a categorical variable (never smoked; former smokers; current smokers) [26].

Child’s Weight Category: Measured as a categorical variable. Calculated using the World Health Organization (WHO) criteria. Underweight: BMI is less than or equal to −2; Normal: BMI is greater than −2 and less than 1; Overweight: BMI is greater than or equal to 1 and less than 2; Obese: BMI is greater than or equal to 2 [27].

### 2.6. Statistical Analysis

Descriptive statistics (means and standard deviations) were reported. Unpaired *t*-tests were used to compare the difference in means among the two groups. One-way analysis of variance (ANOVA) was used to analyze differences in means among three or more groups. CFSS-DS was used as a count outcome and thus a negative binomial regression analysis was conducted. We performed an adjusted negative binomial regression analysis to examine the association between OHRQoL (COHIP-SF19) and children’s dental fear (CFSS-DS). An adjusted negative binomial regression analysis was used to examine the association between children’s dental fear (CFSS-DS) and parental fear. For both negative binomial regression models, we reported the coefficient and 95% confidence interval (95% CI). We selected the negative binomial regression since the distribution of these dependent count variables exhibited over-dispersion. A *p*-value of 0.05 was used to evaluate the significance of all the models in the study. We used Stata/SE16.1 (StataCorp, College Station, TX, USA) to analyze the data.

## 3. Results

Table 1 summarizes the socio-demographic characteristics of children aged 7–12 years and their parents who completed the surveys and the clinical examination. Overall, 93 participants have completed both surveys and clinical examinations. There was an almost equal number of male (53.8%) and female children (46.2%) who participated in the study. A higher child’s age was significantly associated with a higher CFSS score. A total of 32.3% of the children were either overweight or obese. A higher caries percentage was significantly associated with a higher COHIP-SF19 score (poorer OHRQoL) and a higher CFSS-DS score (higher dental fear in children). Figure 1 also shows that children who were accompanied by their mothers had a higher CFSS-DS score than those who were accompanied by their fathers. This value was tested using an independent *t*-test and the difference between them was statistically significant (*p* = 0.021).

Overall, our multivariate analyses (Table 2) showed that with every one-unit increase in children’s level of dental fear (CFSS-DS) their OHRQoL decreased by 0.013 (95% CI: 0.001–0.026) and this value was statistically significant (*p* = 0.037). Further, children with high dental fear (CFSS-DS ≥ 38) had a lower OHRQoL (Coef.: 0.442; 95% CI: 0.050–0.835). This is depicted using a box plot in Figure 2. Lower OHRQoL was also associated with the percentage of caries, in which for every one unit increase in the percentage of caries, the OHRQoL worsened by 0.014 (95% CI: 0.003–0.025) and this value was statistically significant (*p* = 0.013). Finally, after adjusting for various factors such as the child’s age, sex, presence of any medical conditions, parents’ education level, parents’ smoking status, family income, patient’s weight category, presence of malocclusion, and caries percentage, no significant association was found between parental fear of dentists and poorer OHRQoL. However, this lack of association might be influenced by other unmeasured variables, or it may suggest that the effect of parental fear on children’s OHRQoL is more indirect or nuanced than initially hypothesized.

Table 3 depicts the negative binomial regression analyses on children’s level of dental fear by parental levels of dental fears and previous experiences. With every one-unit increase in the parental fear of dentists, the CFSS-DS score increased by 0.185 (95% CI: 0.146–0.224) and this value was statistically significant (*p* < 0.001). Figure 3 depicts a boxplot showing how CFSS-DS score increases gradually as the parent’s fear of dentists increases. Finally, parents who answered that they were anxious on their last visit to the dentist had children with higher levels of dental fear (Coef.: 0.217; 95% CI: 0.068–0.367). 

## 4. Discussion

### 4.1. Dental Fear and Oral Health-Related Quality of Life (OHRQoL)

The findings of our study reveal a significant association between children’s dental fear and their Oral Health-Related Quality of Life (OHRQoL) in Saudi Arabia, a population that has been underrepresented in previous research on dental fear and OHRQoL. By examining these associations within the cultural and healthcare context of Saudi Arabia, our study provides valuable insights into the unique factors influencing pediatric dental fear and oral health outcomes in this population. Consistent with previous research [28,29,30,31,32,33], our results underscore the detrimental impact of dental fear on children’s well-being, particularly in the context of oral health. As children’s fear levels increase, their OHRQoL significantly decreases, highlighting the importance of addressing dental fear in pediatric dental care. 

In our study, we employed a comprehensive approach to assess both children’s dental fear and Oral Health-Related Quality of Life (OHRQoL) using validated scales, namely the Child Fear Survey Schedule—Dental Subscale (CFSS-DS) and the Child Oral Health Impact Profile-Short Form 19 (COHIP-SF19), respectively. This dual assessment enabled a more nuanced understanding of the relationship between dental fear and OHRQoL, providing robust evidence for the observed associations. Our study findings also revealed that a higher caries percentage was significantly associated with a higher COHIP-SF19 score (poor OHRQoL) and a higher CFSS-DS score (higher dental fear in children). These results corroborate the results of previous studies which have concluded a similar positive correlation [16,34,35].

Furthermore, our study utilized a multivariable regression analysis to adjust for potential confounding variables, including demographic, socioeconomic, and clinical factors, such as children’s age, sex, parental education, family income, and dental caries status. By accounting for these variables, we aimed to elucidate the independent contribution of dental fear to OHRQoL while controlling for potential sources of bias and confounding, thus enhancing the reliability and validity of our findings.

### 4.2. Parental Influence on Children’s Dental Fear

Our study also sheds light on the influential role of parental factors, particularly parental dental fear, in shaping children’s dental fear levels. In our study, we observed that increased parental dental fear had a significant impact on increased CFSS-DS scores, leading to poorer OHRQoL. The positive correlation observed in our study between parental dental fear and children’s fear underscores the importance of parental attitudes and behaviors in the transmission of dental fear to their offspring. These findings emphasize the need for targeted interventions aimed at alleviating parental dental fear to mitigate its adverse effects on children’s dental experiences and overall well-being.

Our study results also showed that children who were accompanied by their mothers had a higher CFSS-DS score than those who were accompanied by their fathers. Previous studies also have shown a correlation between a mother’s attitude toward oral care and her child’s oral health status and dental care utilization. Specifically, mothers with high dental care-related anxiety or fear tend to have children with higher levels of caries and fewer dental visits [33].

In another study of 305 children, the authors concluded that children’s dental anxiety was significantly influenced by maternal dental anxiety, post-treatment complications experienced by the mother, and the oral health status of the mother [36].

Potential interventions to reduce dental fear in parents and children include educational programs (parental education, behavioral strategies, informational materials), therapeutic approaches (Cognitive Behavior Therapy (CBT), exposure therapy, family-based interventions), and combined parent–child sessions or VR simulations. The authors of another study concluded that parents and children had positive experiences of CBT and its outcomes and were able to benefit from this psychological treatment when dealing with dental anxiety [37]. Recently, in another study, the authors concluded that Internet-based CBT seems to be an effective treatment for dental and injection phobia in children and adolescents. It reduced fear and anxiety and enabled participants to willingly receive dental treatment [38].

Furthermore, the association between parental fear during dental visits and children’s fear highlights the potential for parental behavior to directly impact children’s perceptions of dental care settings. Interventions targeting parental dental fear management and enhancing parental support and reassurance during dental visits may serve as effective strategies for reducing children’s fear and improving their dental experiences.

### 4.3. Implications for Clinical Practice and Public Health

The implications of our findings extend beyond the research setting and have significant implications for clinical practice and public health initiatives aimed at promoting pediatric oral health. Healthcare professionals, including pediatric dentists and primary care providers, should prioritize the early identification and management of dental fear in children to mitigate its adverse consequences on oral health outcomes and overall quality of life.

Moreover, efforts to enhance parental education and support regarding dental fear and effective coping strategies can empower parents to play a proactive role in fostering positive dental attitudes and behaviors in their children. By fostering a supportive and reassuring dental environment and employing evidence-based interventions, clinicians can contribute to improving children’s dental experiences and promoting lifelong oral health.

### 4.4. Limitations and Future Directions

While our study provides valuable insights into the association between dental fear and OHRQoL among Saudi children, several limitations warrant consideration. The cross-sectional design precludes causal inferences regarding the observed relationships, highlighting the need for longitudinal studies to elucidate the temporal dynamics of dental fear and its impact on oral health outcomes over time.

Additionally, the reliance on self-reported measures of dental fear and OHRQoL introduces the potential for response bias and measurement error, necessitating caution in the interpretation of findings. Future research employing objective clinical assessments and longitudinal follow-up is warranted to corroborate and extend our findings.

Furthermore, the study’s sample primarily comprised children and parents from a single healthcare setting, limiting the generalizability of findings to broader populations. Future studies incorporating diverse samples and settings are needed to enhance the external validity of findings and inform culturally sensitive interventions tailored to specific populations.

## 5. Conclusions

Our study contributes to the growing body of evidence elucidating the complex interplay between dental fear, parental factors, and oral health outcomes among Saudi Arabian children. By addressing these factors through targeted interventions and comprehensive dental care approaches, especially cognitive behavior therapy and combined parent-child sessions, healthcare providers can foster positive dental attitudes and behaviors, ultimately enhancing the oral health and well-being of pediatric populations.

## Figures and Tables

**Figure 1 ijerph-21-01195-f001:**
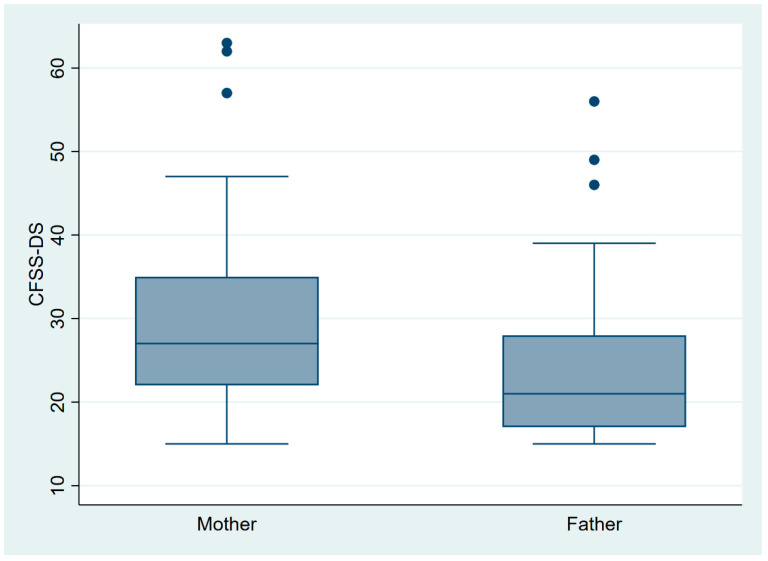
The association between parents and their child’s level of fear.

**Figure 2 ijerph-21-01195-f002:**
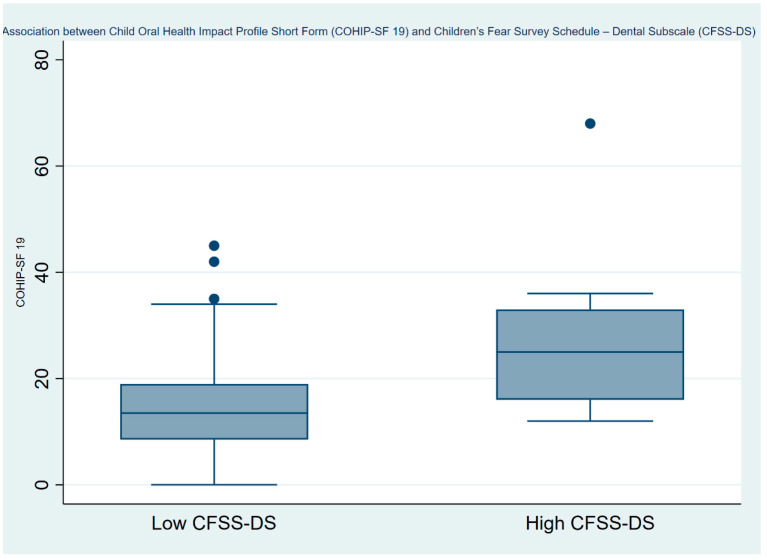
The association between the Child Oral Health Impact Profile Short Form (COHIP-SF19) and the Children’s Fear Survey Schedule -—Dental Subscale (CFSS-DS).

**Figure 3 ijerph-21-01195-f003:**
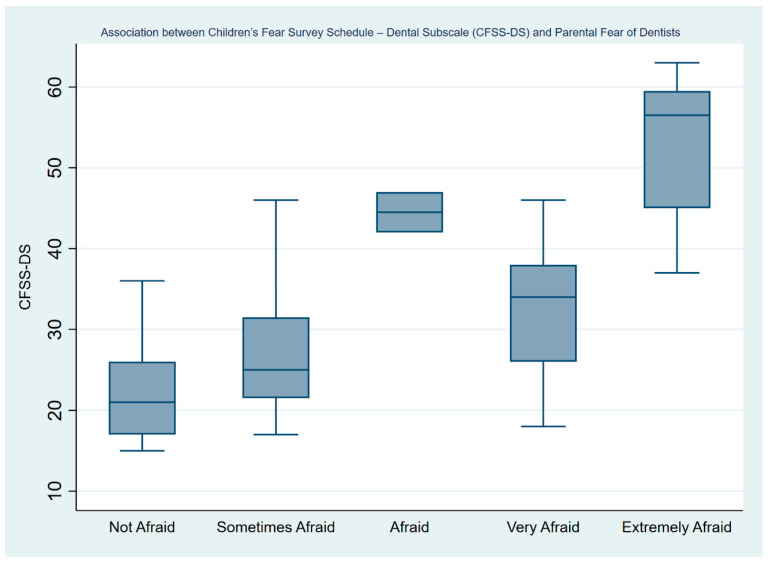
The association between the Children’s Fear Survey Schedule—Dental Subscale (CFSS-DS) and parental fear of dentists.

**Table 1 ijerph-21-01195-t001:** Demographics of Saudi children between the ages of 7 and 12 years and their parents who have completed the Child Oral Health Impact Profile Short Form (COHIP-SF 19) and Children’s Fear Survey Schedule—Dental Subscale (CFSS-DS). N = 93.

	Overall Number (%)	COHIP-SF19 (SD)	*p*-Value *	CFSS-DS (SD)	*p*-Value *
Total	93 (100%)	16.48 (11.10)		27.40 (11.33)	
Age (mean)	9	--	0.754	--	0.020 *
Sex					
*Male*	50 (53.8%)	17.30 (9.03)	0.447	25.56 (10.79)	0.092
*Female*	43 (46.2%)	15.53 (13.15)		29.53 (11.69)	
Child Medical Conditions **					
*Yes*	4 (4.3%)	16.78 (11.23)	0.217	27.78 (11.38)	0.130
*No*	89 (95.7%)	9.75 (4.11)		19.00 (6.16)	
Parental Education					
*Less than high school*	8 (8.6%)	12.5 (11.40)	0.569	29.38 (12.60)	0.529
*High school*	27 (29.0%)	16.63 (10.66)		29.04 (14.24)	
*More than high school*	58 (62.4%)	16.97 (11.80)		26.36 (9.61)	
Parental Smoker					
*Never*	76 (81.7%)	15.78 (9.86)	0.050	27.17 (11.14)	0.357
*Current*	15 (16.1%)	21.67 (15.28)		29.80 (12.62)	
*Former*	2 (2.2%)	4.5 (6.36)		18.00 (0.00)	
Family Income †					
*Low*	31 (33.3%)	17.26 (10.63)	0.403	27.84 (11.64)	0.141
*Medium*	19 (20.4%)	18.79 (12.30)		31.42 (14.40)	
*High*	43 (46.2%)	14.91 (10.90)		25.30 (9.13)	
Child Weight Category ‡					
*Underweight*	6 (6.5%)	10.17 (6.74)	0.449	24.67 (6.50)	0.836
*Normal*	57 (61.3%)	16.40 (9.92)		27.05 (12.09)	
*Overweight*	12 (12.9%)	16.67 (11.58)		27.67 (10.23)	
*Obese*	18 (19.4%)	18.72 (14.93)		29.22 (11.24)	
Caries Percentage §	13.45	--	0.008 *	--	0.001 *

SD—Standard Deviation; COHIP-SF 19—Child Oral Health Impact Profile Short Form; CFSS-DS—Children’s Fear Survey Schedule—Dental Subscale; * When groups contained only two categories an independent *t*-test was performed. When groups contained three or more categories a one-way analysis of variance (ANOVA) was performed. When the independent variable was continuous (age/caries percentage) a univariate negative binomial regression was used; *p*-value < 0.05 indicates a statistically significant difference; ** Child medical conditions consisted of the presence of any chronic medical condition. Special needs patients were, however, excluded from the study; † Family income was defined according to the King Khaled Foundation for determining the poverty line in Saudi Arabia; ‡ Calculated using the World Health Organization (WHO) criteria. Underweight—BMIz is less than or equal to −2, Normal—BMIz is greater than −2 and less than 1, Overweight—BMIz is greater than or equal to 1 and less than 2, Obese—BMIz is greater than or equal to 2; § Percentage of caries was calculated as the total number of carious primary and permanent teeth divided by the total number of primary and permanent teeth.

**Table 2 ijerph-21-01195-t002:** Negative binomial regression analysis studying the association between the Child Oral Health Impact Profile Short Form (COHIP-SF 19) with the Children’s Fear Survey Schedule—Dental Subscale (CFSS-DS), percentage of caries, and parental fear of dentists after adjusting for child’s age, sex, presence of any medical conditions, parents’ education level, parents smoking status, family income, patient’s weight category, presence of malocclusion, and caries percentage. (N = 93).

Exposure/Outcome	COHIP-SF19		
	Adjusted Coef.	95% CI	*p*-Value
CFSS-DS (Continuous)	0.013	0.001–0.026	0.037 *
CFSS-DS (Reference: Low) **	0.442	0.050–0.835	0.027 *
Percentage of Caries †	0.014	0.003–0.025	0.013 *
Parental Fear of Dentist ‡	0.097	−0.006–0.200	0.064
Parent’s Previous Bad Experience ¶	0.069	−0.291–0.429	0.708
Parent’s Last Visit §	0.174	−0.103–0.450	0.219

95% CI: 95% Confidence Interval; COHIP-SF 19—Child Oral Health Impact Profile Short Form; CFSS-DS—Children’s Fear Survey Schedule—Dental Subscale; Coef.—Coefficient; * *p*-value < 0.05 and this value is statistically significant; ** Analyzed as a binary variable. CFSS ≥ 38 = High; † Percentage of caries was calculated as the number of carious primary and permanent teeth divided by the total number of primary and permanent teeth. Adjusted for child’s age, sex, presence of any medical conditions, parents’ education level, parents smoking status, family income, patient’s weight category, presence of malocclusion, and child’s anxiety (CFSS-DS); ‡ Measured using a 5-point Likert scale. Analyzed as a continuous variable. ¶ Measured as a binary variable where they answered “yes/no” to “if they had ever had a previous bad experience when they were a child with a dentist”; § Measured as a binary variable where they answered, “comfortable or anxious or higher” to “how was your last experience with your visit to the dentist”.

**Table 3 ijerph-21-01195-t003:** Negative binomial regression analysis studying the association between the Children’s Fear Survey Schedule—Dental Subscale (CFSS-DS) with parental fear of dentists after adjusting for child’s age, sex, presence of any medical conditions, parents’ education level, parents’ smoking status, family income, patient’s weight category, presence of malocclusion, and caries percentage (N = 93).

Exposure/Outcome	CFSS-DS		
	Adjusted Coef.	95% CI	*p*-Value
Parental Fear of Dentists †	0.185	0.146–0.224	<0.001
Parent’s Previous Bad Experience ‡	0.126	−0.067–0.319	0.202
Parent’s Last Visit ¶	0.217	0.068–0.367	0.004 *

95% CI: 95% Confidence interval; CFSS-DS—Children’s Fear Survey Schedule—Dental Subscale; * *p*-value < 0.05 and this value is statistically significant; † Measured using a 5-point Likert scale. Analyzed as a continuous variable. ‡ Analyzed as a binary variable where they answered “yes/no” to “if they had ever had a previous bad experience with a dentist when they were a child”. ¶ Analyzed as a binary variable where they answered, “comfortable or anxious or higher” to “How was your last experience with your visit to the dentist”.

## Data Availability

The datasets used and analyzed during the current study are available from the corresponding author on reasonable request.

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
