# Peer review of "Association between Dental Fear and Children’s Oral Health-Related Quality of Life"

_ijerph, 2024, doi:10.3390/ijerph21091195_

Round 1

Reviewer 1 Report

Comments and Suggestions for Authors

I would like to express my deep gratitude to the authors for the significant contribution that this study makes to understanding dental fear in children and its influence on oral health-related quality of life (OHRQoL). This work not only explores a highly relevant clinical and social issue but does so in a specific and under-studied context like Saudi Arabia, thereby enriching the scientific literature with data that can guide future research and practical interventions. The choice of validated measurement tools and the rigorous statistical analysis demonstrate meticulous attention to detail and a solid methodological approach. This study is an excellent example of how research can have a significant impact on clinical practice and public health.

Strengths of the Article

  1. Relevance of the Topic: The article addresses a highly important issue, namely the association between dental fear in children and their oral health-related quality of life (OHRQoL). This relationship is crucial for improving the management of pediatric patients in dental settings.

  2. Specific Cultural Context: The study focuses on a specific population, Saudi children, contributing data that fill a gap in the existing literature. This is particularly useful for developing targeted interventions in that region.

  3. Use of Validated Tools: The use of validated questionnaires in Arabic, such as the COHIP-SF19 and CFSS-DS, ensures the validity and reliability of the collected data, providing a solid foundation for the study’s conclusions.

  4. Rigorous Statistical Analysis: The adoption of negative binomial and logistic regression models to analyze the data demonstrates a robust methodological approach, accounting for potential over-dispersion and the presence of confounding variables.

  5. Clinical and Public Health Implications: The study’s results have clear implications for clinical practice and public health policies, suggesting the need for targeted interventions to reduce dental fear in children, thereby improving their quality of life and long-term oral health.

Recommended Modifications

Abstract

  • Line 25-26: "the odds of having high dental fear in children increased by 12.97 (95%CI: 1.29-130.77) with every one-unit increase in parental dental fear."
    • Suggestion: The confidence interval is very wide, which might indicate significant variability in the data or a limited sample size. It may be useful to briefly discuss this aspect in the abstract’s limitations or consider supplementary analysis to reduce the interval width.

Introduction

  • Line 57-58: "previous studies conducted in Saudi Arabia did not adequately account for socio-demographic and clinical factors that could have acted as confounders."
    • Suggestion: Specify which studies did not consider these factors and explain exactly how the present study addresses these limitations. This might include citing specific studies and providing a more detailed description of the shortcomings your study has addressed.

Methods

  • Line 75: "Sample size calculation was done."

    • Suggestion: Rephrase this sentence in a more academic manner. For example: "A sample size calculation was conducted to ensure adequate statistical power for the study."
  • Line 82-83: "good health" and "special care needs or children with a debilitating medical condition."

    • Suggestion: Define more precisely what is meant by "good health" and which specific medical conditions led to the exclusion of children from the study. This will help clarify the inclusion/exclusion criteria and improve the replicability of the study.
  • Line 86: "Informed consent was obtained from parents as part of our online questionnaire."

    • Suggestion: Add details on how complete understanding of the informed consent was ensured for parents, considering that the questionnaire was administered online. For example, mention if additional explanations were provided or if there was an opportunity to ask questions.

Results

  • Line 164-165: "children who were accompanied by their mothers had a higher CFSS-DS score than those who were accompanied by their fathers."

    • Suggestion: Further interpret this data point. It might be useful to explore why the presence of the mother is associated with higher levels of fear and discuss whether this aligns with other studies. Adding a comment on potential cultural or psychological factors that could explain this difference would be beneficial.
  • Line 176: "Finally, parental fear of dentists was not associated with poorer OHRQoL after adjusting for..."

    • Suggestion: This statement could be improved by clarifying that the lack of association might be due to other factors not considered, or that the effect might be less direct than hypothesized. It might be helpful to suggest further research to explore these aspects.

Discussion

  • Line 272-273: "the positive correlation observed in our study between parental dental fear and children's fear underscores the importance of parental attitudes and behaviors..."
    • Suggestion: Consider adding a paragraph discussing potential interventions to reduce dental fear in parents, which in turn could positively influence their children. This might include suggestions for educational programs or therapeutic approaches that address dental fear at the family level.

Conclusions

  • Line 312-317: "By addressing these factors through targeted interventions and comprehensive dental care approaches, healthcare providers can foster positive dental attitudes and behaviors, ultimately enhancing the oral health and well-being of pediatric populations."
    • Suggestion: The conclusions could be further strengthened by specifying which types of interventions would be most effective in this context. It might be useful to suggest practical guidelines for clinicians or specific proposals for public health policies.

Other Sections

  • References: Ensure that all citations are up-to-date and relevant. Some sources could be supplemented with more recent research to further enrich the discussion and evidence base

I would like to suggest the inclusion of the following article in your literature review, as I believe it can enrich and strengthen the theoretical and methodological context of your study:

Diotaiuti, P., Corrado, S., Mancone, S., Cavicchiolo, E., Chirico, A., Siqueira, T. C., & Andrade, A. (2023). A psychometric evaluation of the Italian short version of the Fear of Pain Questionnaire-III: Psychometric properties, measurement invariance across gender, convergent, and discriminant validity. Frontiers in Psychology, 13, 1087055. https://doi.org/10.3389/fpsyg.2022.1087055

How and Where to Include It

  1. Introduction (Section on the Measurement of Dental Fear): You might cite this article when discussing methodologies for measuring fear, especially regarding the validity and reliability of psychometric tools used in the context of dental fear. Including this reference could strengthen your argument on the importance of using validated tools, emphasizing the need to consider convergent and discriminant validity, as discussed by Diotaiuti et al. (2023).

    Suggested Integration: "The importance of using validated instruments to assess dental fear is emphasized by studies such as Diotaiuti et al. (2023), which highlighted the psychometric properties and measurement invariance of fear-related questionnaires. This approach ensures that the tools used are both reliable and applicable across different populations, including the assessment of gender differences in fear responses."

  2. Discussion (Section on Validity and Reliability of Instruments): When analyzing the results, it may be useful to cite the article to support the discussion on the validity and reliability of the psychometric tools used in your study. You could refer to the evidence from Diotaiuti et al. (2023) to better contextualize the importance of ensuring that the tools used are valid across various demographic variables, such as gender.

    Suggested Integration: "The findings regarding the reliability of the CFSS-DS in our study align with recent psychometric evaluations in related fields. For instance, Diotaiuti et al. (2023) demonstrated the importance of ensuring measurement invariance across gender in fear assessment tools, a factor that could be further explored in future studies on dental fear."

  3. Methods (Section on Psychometric Instruments): When describing the tools used to measure dental fear in children, it might be useful to mention this article to support the choice and validation of the tool, emphasizing the importance of assessing psychometric properties similarly to what was done by Diotaiuti et al. (2023).

    Suggested Integration: "In line with the approach taken by Diotaiuti et al. (2023) in their evaluation of the Fear of Pain Questionnaire-III, our study employed psychometrically validated instruments to ensure the robustness of the data collected, particularly in terms of convergent and discriminant validity."

  •  

Reviewer 2 Report

Comments and Suggestions for Authors

The manuscript as such is well written and the research study is sound. I would  suggest authors to put the inclusion criteria in a tabulated format with more specifics along with exclusion criteria. The references are not in one uniform  format and need to be corrected. 

Reviewer 3 Report

Comments and Suggestions for Authors

Thank you for the opportunity to review this manuscript. The manuscript “Association between dental fear and children’s oral health related quality of life” presents a study that investigates the association between children’s fear levels and oral health-related quality of life in Saudi Arabia and also studies the association between parental dental fear and children’s fear. Considering the broader context of presented studies and further impact of the published papers, I have few corrections /suggestions in the introduction and methodology section, suggested to be addressed before next steps in the publication.

1.      The introduction section does not include information about oral hygiene practice and oral hygiene awareness followed in Saudi Arabia in general and in the study area (Eastern Province, Dammam) in particular.

2.      I would suggest including information about fluoride levels in drinking water of the region and comparatives with World Health Organization reference levels, so that the readers have reference values.

3.      The authors have mentioned education and have included as one of the main confounders, was it children’s education/grades? OR parents’ education. It would be important to mention parents’ education separately for both of the parents as the results are mostly affected by mothers’ education and to a lesser extent by fathers’ education.

4.      One important confounding factor is oral hygiene practice, was it made sure to include only individuals with similar oral hygiene practice so that the issue of selection bias is avoided?

5.      The study population included children aged 07-12 years, it would have been important to have obtained both assent from children and inform consent from the parents; as inform consent from the parents is obtained, children assent also needs to be confirmed by a statement in the first part of the methodology section.

6.      In the statistical analysis section, the authors have exclusively used mean as a descriptive measure, did the authors confirm the data normality before the statistical analysis?

Minor comments:

1.      I would suggest replacing “Oral health-related quality of life” and “dental fear” in key words with suitable similar words. As these words have already been used in the title of the manuscript.

Comments on the Quality of English Language

NA

Round 2

Reviewer 3 Report

Comments and Suggestions for Authors

Thank you for the opportunity to review the revised version of the manuscript “Association between dental fear and children’s oral health related quality of life”. I have no further comments. I wish best of luck to all the authors.